**Intercomparison of Fast airborne Ozone Instruments to measure Eddy Covariance Fluxes:**

**Spatial variability in deposition at the ocean surface and evidence for cloud processing**

Randall Chiu[1,2], Florian Obersteiner[3], Alessandro Franchin[4], Teresa Campos[4], Adriana Bailey[4],

5  Christopher Webster[4], Andreas Zahn[3], and Rainer Volkamer[1,2,*]

1 Department of Chemistry, University of Colorado Boulder, 215 UCB, Boulder, CO, USA

2 Cooperative Institute for Research in Environmental Sciences (CIRES), University of Colorado
Boulder, 216 UCB, Boulder, CO, USA

3 Karlsruhe Institute of Technology, Institute of Meteorology and Climate Research (IMK),

10  Dept. ASF, Hermann-von-Helmoltz-Platz 1 D-76344 Eggenstine-Leopoldshafen, Germany

National Center for Atmospheric Research, P.O. Box 3000, Boulder, CO, 80307, USA

* corresponding authors: randall.chiu@colorado.edu, rainer.volkamer@colorado.edu

## Abstract

The air-sea exchange of ozone is controlled by chemistry involving halogens, dissolved organic

carbon and sulfur in the sea surface microlayer. Calculations also indicate faster ozone photolysis

at aqueous surfaces, but the role of clouds as ozone sink is currently not well established. Fast

response ozone sensors offer opportunities to measure eddy covariance (EC) ozone fluxes in the

marine boundary layer. However, intercomparisons of fast airborne $O_3$ sensors, and EC $O_3$ fluxes

measured on aircraft have not been conducted before. In April 2022, the TI³GER (Technical



Innovation Into Iodine and GV aircraft Environmental Research) field campaign deployed three fast ozone sensors (gas chemiluminescence and a combination of UV absorption with coumarin chemiluminescence detection, CID) together with a fast water vapor sensor and anemometer to study iodine chemistry in the troposphere and stratosphere over Colorado and over the Pacific

Ocean near Hawaii and Alaska. Here, we present an instrument comparison between the NCAR Fast $O_3$ instrument ($FO_3$, gas-phase CID) and two KIT Fast AIRborne Ozone instruments (FAIRO, UV absorption and coumarin CID). The sensors have comparable precision <0.4% $Hz^{-0.5}$ (0.15 ppbv $Hz^{-0.5}$), and ozone volume mixing ratios (vmr) generally agreed within 2% over a wide range of environmental conditions: $10 < O_3 < 1000$ ppbv; below detection $< NO_x < 7$ ppbv;

and 2 ppmv $< H_2O < 4\%$ VMR. Both instrument designs are demonstrated to be suitable for EC flux measurements and were able to detect $O_3$ fluxes with exchange velocities (defined as positive for upward) as slow as $-0.010 \pm 0.004$ cm $s^{-1}$, which is in the lower range of previously reported measurements. Additionally, we present two case studies: one in which the direction of ozone and water vapor fluxes were reversed ($v_{O3} = +0.134 \pm 0.005$ cm $s^{-1}$), suggesting that

overhead evaporating clouds could be a strong ozone sink; and another in which ozone fluxes $v_{O3}$ are negative (varying by a factor of 6-10 from $-0.036 \pm 0.006$ to $-0.003 \pm 0.004$ cm $s^{-1}$), while the water vapor fluxes are consistently positive due to evaporation from the ocean surface and spatially homogeneous. Future work is needed to better understand the role of clouds as a possibly widespread sink of ozone in the remote marine boundary layer, and to elucidate possible

drivers (physical, chemical, or biological) of the variability in ozone exchange velocities on fine spatial scales (~20 km) over remote oceans.



## 1. Introduction

In the troposphere, ozone is a pollutant with adverse health effects for both animals and plants. Eddy covariance (EC) is a technique that has been commonly employed to determine the fluxes

of ozone to terrestrial and marine ecosystems. In terrestrial environments, EC flux measurements have been made over periods of months to years (Bauer et al., 2000; Güsten and Heinrich, 1996). Over land, uptake to soils and plant stomata are the major sink of ozone (Clifton et al., 2020; Massman et al., 1995). Consequently, previous campaigns have measured ozone fluxes over a variety of terrestrial settings including agricultural lands (Stella et al., 2011; Zhu et al., 2014;

Lamaud et al., 2009; Zhu et al., 2015; Massman et al., 1995; Zhu et al., 2020), forests (Juráň et al., 2019; Finco et al., 2017; Zeller, 2002; Lamaud et al., 2002; Zeller and Nikolov, 2000; Kammer et al., 2019; Fares et al., 2014; Vermeuel et al., 2021; Altimir et al., 2006; Rannik et al., 2012), grasslands (Muller et al., 2009; Wohlfahrt et al., 2009), peatlands (El-Madany et al., 2017), and deserts (Güsten et al., 1996).

Oceans account for ~1/3 of global ozone dry deposition (Ganzeveld et al., 2009). Ozone losses in the marine environment may be driven by reactions with halogens such as iodide (Saiz-Lopez et al., 2012) or with double bonds from fatty acid precursors (Chiu et al., 2017). EC flux measurements of ozone have also been performed in coastal and oceanic settings (Bariteau et al., 2010; Helmig et al., 2006; Gallagher et al., 2001) and over sea ice (Barten et al., 2023; Muller et

al., 2012).

Whereas EC flux measurements of ozone are numerous, comparison studies are fewer. Ozone fluxes from EC methods have been compared to those from gradient measurements (Muller et al., 2009; Zhu et al., 2020; Loubet et al., 2013) and dynamic chamber methods (Plake et al.,





2015). However, comparisons of co-located EC flux measurements are uncommon. Aircraft

measurements have been performed near tower facilities (Massman et al., 1995), but ozone

fluxes at altitude may differ from those at ground level. To our knowledge, the only aircraft

instrument intercomparison for ozone EC flux was performed by Muller et al., (2010), who

measured ozone EC flux using two dry chemiluminescence instruments over grassland. The

PASE campaign flew two ozone instruments over the equatorial Pacific Ocean, but only one had

fast response suitable for EC flux calculation (Conley et al., 2011). Here we present the first

aircraft ozone EC flux intercomparison of three ozone instruments of two different designs over

remote marine air.

Section 2 introduces the Technological Innovation Into Iodine and GV Environmental Research

(TI3GER) field campaign, and describes the instruments and methods used to calculate fluxes of

$O_3$ and $H_2O$ by the EC technique. Section 3 compares the $O_3$ concentrations and EC fluxes in

context with the available literature over oceans, assesses spatial variability and the EC flux error

budget. Finally, Section 4 summarizes the conclusions and gives an outlook for future work.

## 2. Methods

### 2.1. The TI³GER field campaign

In April 2022, the Technological Innovation Into Iodine and GV Environmental Research

(TI³GER) technical campaign was performed to lay the groundwork for future field

investigations into the interactions of ozone and iodine in the upper troposphere lower

stratosphere (UTLS). In total, eight research flights (RFs) were conducted, with RFs 01 and 02

over the continental United States, and RFs 03-08 conducted over the Pacific Ocean near Hawaii

and Alaska. Among the instruments flown on TI3GER were three ozone instruments, two of



which were of an identical design. The NCAR Fast O3 instrument operates by $NO_2$ chemiluminescence and has been in use since the early 1970s (Ridley et al., 1992, 1972; Ridley and Howlett, 1974; Pearson, R. and Stedman, 1980). Two copies of the Fast AIRborne Ozone (FAIRO) instrument from KIT were also deployed (FAIRO 1 and FAIRO 2). The FAIRO

instruments operate by coumarin chemiluminescence calibrated against a dual-beam UV absorption photometer.

One objective of TI³GER was to compare the performance of the two instrument designs and evaluate their ability to measure EC flux of ozone on the NCAR/NSF Gulfstream 5 (GV) platform. The GV measures 3-D winds using a combination of measurements from pitot, static,

and radome sensors; the vertical components of these 3-D winds are needed for EC analysis. In all, EC flux measurements were performed during nineteen legs flown over the Pacific Ocean. The continental flights are not discussed here because they did not include many EC flux measurements. A table of the relevant meteorological and ocean-state variables appears in the supplement as Table S1. Figure 1 shows a map of where attempts were made to measure EC flux.

The arrows point to the locations of the flux legs with curves showing the five-day back-trajectories of the sampled air calculated by HYSPLIT (Stein et al., 2015; Rolph et al., 2017) using the National Oceanographic and Atmospheric Administration (NOAA) National Centers for Environmental Prediction (NCEP) Global Forecasting System (GFS) meteorological dataset.

### 2.2. Ozone Instrumentation

Three ozone instruments were installed on the GV. Two (the FAIRO instruments) were of an identical design.





### 2.2.1. The NCAR Fast O3 instrument

The NCAR Fast O3 instrument sampled from a HIAPER Modular Inlet (HIMIL). All tubing was made of Teflon. The total mass flow in the inlet was 2370 sccm. The sample line was 70 cm long

with an inner diameter of 6.4 mm. From this flow, Fast O3 sampled 500 sccm through a 140 cm-long line with an inner diameter of 3.8 mm. All flows had a constant absolute pressure of 70 torr. Fast O3 provides 10 Hz data by detecting photons from the following chemiluminescence reaction:

$$O_3 + NO \rightarrow O_2 + NO_2^* \tag{R1}$$

$$NO_2^* \rightarrow NO_2 + h\nu \tag{R2}$$

The excited $NO_2$ in R2 can also be quenched by collision with other molecules. Water vapor quenches excited $NO_2$ more efficiently than do nitrogen or oxygen (Matthews et al., 1977), so after time stamp synchronization among the instruments (see Sect. 2.3.), the following water vapor correction is applied (Ridley et al., 1992):

$$[O_3]_{corrected} = [O_3] \times (1 + 4.3(\pm 0.3) \times 10^{-3} \times [H_2O]) \tag{1}$$

where $[O_3]$ is the ozone mixing ratio in ppbv, and $[H_2O]$ is the water vapor mixing ratio in permille by volume of dry air.

The water vapor correction is performed using VCSEL water vapor data (see Sect. 2.2.) that are collected at a higher frequency (25 Hz) than are the Fast O3 data. Thus, the water vapor

correction is expected to contribute negligible bias to the EC flux calculations. To assess the potential impact of the water vapor correction on Fast O3 EC fluxes, the constant in eq. (1) was varied from its minimum and maximum estimated values (4.0 – 4.6) in the RF03-C-2 leg; the





change in this parameter resulted in biases in the EC flux results of no more than 0.7%.

Neglecting the water vapor correction altogether decreased the calculated exchange velocity (see

Sect. 2.4) by 5% from 0.131 cm s$^{-1}$ to 0.124 cm s$^{-1}$ (see Table 2). However, since the 0.131 cm s$^{-1}$

value is in good agreement with the EC flux results from FAIRO 1 and FAIRO 2, and since the

Fast O3 exchange velocities are not systematically higher or lower than those from FAIRO 1 or

FAIRO 2, we believe that the water vapor correction is not a source of bias in EC flux

calculations.

The Fast O3 instrument was calibrated after the campaign using a TECO Model 49i-PS ozone

primary standard. The typical instrument detection limit is 0.5 ppbv Hz$^{-0.5}$ with an accuracy not

better than 5% at high signal to noise.

### 2.2.2. The KIT Fast AIRborne Ozone (FAIRO) instruments

Two identical FAIRO instruments were deployed. The FAIRO instruments were independently

checked for proper functioning both prior to the campaign using an Ansyco (now Gasmet

Technologies) SYCOS KT-O3M and after the campaign using a TECO 49i-PS. The FAIROs

sampled from a separate HIMIL (aft-facing inlet line) through a PFA line with a length of 4.3 m

and a 1/6-inch inner diameter. Outside air was pulled at 11 vol.-L min$^{-1}$ at ambient pressure by a

Vacuubrand MD1 pump downstream of the instruments. The flow was split at a T-fitting ~0.5 m

ahead of the FAIROs. Internally, 2.5 vol.-L min$^{-1}$ of flow went to the UV photometer, which

measured ozone absorption around 255 nm within the Hartley band. The $O_3$ absorption cross

section and temperature dependence are taken from (Barnes and Mauersberger, 1987). The UV

absorption channel operates at 0.25 Hz. A second, faster 12.5 Hz coumarin chemiluminescence

detector (CID) (Ermel et al., 2013) is calibrated against the UV channel and provides the data



used in EC flux calculations. The dual detector FAIRO design has two main advantages over the

Fast O3 instrument: the FAIROs are lightweight (approx. 14 kg, 19'' rack slot with 3 height units

per instrument) and do not require operating fluids such as compressed gases. Scattering by

aerosols and absorption by aromatic compounds and water vapor are well-known interferences

for UV ozone instruments (Dunlea et al., 2006). The potential for humidity changes to

interference with FAIRO uv photometers was further investigated, and is found to be small yet

not fully insignificant (see Supplementary Figure S1). Interference from aerosols is avoided by

the backward-facing sample inlet and aromatic compounds are expected to be minimal in the

pristine air sampled in RFs 03-07. A detailed technical description of FAIRO CID can be found

in (Zahn et al., 2012). The instrument detection limit is below 1 ppbv $Hz^{-0.5}$ (provided by the

CID) and the total uncertainty 1.5% (mainly determined by the uncertainty of the $O_3$ absorption

cross section found in the literature) or 1.5 ppbv, whatever is lower.

### 2.2. Water vapor: VCSEL

Water vapor in the free stream above the GV is measured by the vertical cavity surface emitting

laser (VCSEL) hygrometer. VCSEL is an open-path optical cavity measuring two absorption

lines for high dynamic range: a strong line at 1854.03 nm for low vmrs and a weak line at

1853.37 nm for high vmrs. Data are collected at 25 Hz. During the flux legs, water vapor is

always above the VCSEL detection limit of 0.8 ppmv. Details about the operation of VCSEL can

be found in Zondlo et al., (2010).

### 2.3. Instrument time stamp synchronization

The three ozone instruments and the GV variables are measured on four independent time

stamps, each with its own potential offset and drift. Conveniently, ozone and water vapor vmrs



are occasionally anticorrelated (and less frequently, correlated). We use these anticorrelation

events to synchronize each ozone instrument with VCSEL since VCSEL is already synchronized

to the anemometer. First, the ozone and VCSEL signals are interpolated to a common 100 Hz

timestamp. Second, each ozone time series is visually inspected to identify unambiguous

anticorrelation events with the water vapor time series in the periods before and after each flux

leg. Third, the time lag at each anticorrelation event is determined by shifting the interpolated

ozone signal until the absolute value of the covariance between the ozone and VCSEL signals is

maximized. Finally, with the time lag identified both before and after the leg, the ozone time

stamp is linearly stretched to match the VCSEL time stamp.

## 2.4. Eddy covariance flux calculations

Eddy covariance (EC) is a commonly used technique to determine the fluxes of gases in well-

mixed surface layers. Given chemical concentration and wind speed data, EC flux can be

calculated as:

$$ECflux = cov(x, w) = \frac{1}{n}\sum_{i=1}^{n}(x_i - \overline{x})(w_i - \overline{w}) \qquad (2)$$

where x is the concentration of the chemical species and w is the vertical wind component. For

non-stationary conditions, wavelet analysis (WA) is commonly employed instead (Torrence and

Compo, 1998). In WA, the time series are first transformed into a wavelet by convolution with a

wavelet function:

$$W_s(a, b) = \int_{-\infty}^{\infty} s(t)\,\psi_{a,b}(t)dt \qquad (3)$$

$$\psi_{a,b}(t) = \frac{1}{\sqrt{a}}\psi_0\left(\frac{t-b}{a}\right) \qquad (4)$$





where a and b are translation factors for $\Psi_0$, the "mother wavelet" function. For eddy covariance

applications, the typical choice for the mother wavelet is the Morlet wavelet:

$$\psi_0(\eta) = \pi^{\frac{-1}{4}} e^{i\omega_0\eta} e^{\frac{-\eta^2}{2}}, \omega_0 = 6 \tag{5}$$

The WA flux is then calculated as $|W_w W_x|$, where $W_w$ and $W_x$ are the wavelet coefficients of

wind and ozone, respectively (Wolfe et al., 2018).

A challenge in EC flux error analysis is that EC flux is not a measurement from a single

instrument, but rather the combination of measurements from two instruments: a chemical

monitor of some sort and an anemometer. For individual instruments, estimation of the limit of

detection (LOD) from random error (RE) can be straightforward:

$$LOD = \alpha \times \Re \tag{6}$$

where α is a dimensionless factor corresponding to the confidence level (1.96 for 95% CL, 3 for

99% CL). The standard deviation of blank measurements can be used to estimate the RE of a

single instrument. However, this method is not applicable to flux measurements due to the lack

of true "blanks" matching the chemical and meteorological conditions of interest.

Several methods for determining the LOD of EC flux measurements have been put forth based

on statistical treatments of the cross-covariance of the chemical and wind data at different time

lags. For example, (Langford et al., 2015) present the following formula for estimating the root

mean squared error (RE$_{RMSE}$):

$$\Re_{RMSE} = \sqrt{0.5\left(\left(\sigma_{f_{w'c'_{[-\Gamma]}}}\right)^2 + \left(\overline{f_{w'c'_{[-\Gamma]}}}\right)^2 + \left(\sigma_{f_{w'c'_{[+\Gamma]}}}\right)^2 + \left(\overline{f_{w'c'_{[+\Gamma]}}}\right)^2\right)} \tag{7}$$



where $\sigma_{f_{w'c'}}$ and $\overline{f_{w'c'}}$ are the standard deviation and average of the cross-covariance, and $\pm\Gamma$ represent time lags far away from the true time lag between the wind and chemical measurements. Currently, no well-established method for estimating the LOD of EC and WA fluxes is commonly accepted. The number of independent replicate measurements of ozone

available during TI³GER gave us the unique opportunity to explore, evaluate and optimize methods to constrain the uncertainty of EC fluxes, since the standard deviation of the fluxes measured between the individual instruments can give a sense of the magnitude of the "true" error.

A MATLAB toolkit (AirChem/FluxToolbox: Collections of scripts for eddy covariance flux

calculations (both traditional and wavelet-based)., 2023) was used for this work. Raw data must be pre-processed to remove data gaps before inputting to the toolbox. Data gaps are removed by linear interpolation; such gaps are rare, and interpolation is used only to remove up to three or four points (out of ~2000-4000 which is typical for a flux leg). Because the GV data are recorded at higher resolution than are the ozone data, the wind and VCSEL data are binned to each ozone

instrument's corrected time stamp.

For EC fluxes, the toolbox detrends the data with a boxcar method in a user-defined time frame. The lengths of the detrending time frames were selected to balance being short enough to remove systematic cross-covariance structures with being long enough to retain low-frequency fluxes. A detrending time of 10 s was used in all fluxes presented below. At typical aircraft speeds, 10 s

corresponds to 1-1.2 km. In addition to calculating an eddy covariance flux, the toolbox also calculates WA flux (Torrence and Compo, 1998) and outputs cospectra as a function of frequency.



In contrast to common practice, we express ozone fluxes in terms of exchange velocity ($v_e$) rather than deposition velocity ($v_d$), where:

$$v_e = \frac{flux(ppb\, m\, s^{-1})}{concentration(ppb)} \times \frac{100cm}{1m} \tag{8}$$

Exchange velocity is the same as deposition velocity apart from the lack of a negative sign, i.e. upward directed fluxes have positive $v_e$. We use $v_e$ rather than $v_d$ because some interesting case studies presented have upward directed fluxes, which are more intuitively represented using positive signs.

Notably, fluxes and cross-covariances in principle have the same units (molec $cm^{-2}$ $s^{-1}$ or ppb m $s^{-1}$). However, we use "covariance" to refer to the cross-covariance calculated for different lag times by our code, and "flux" to identify an atmospheric state. This distinction is useful when discussing EC flux errors, which are estimated from cross-covariances at time lags departing from the true lag between instruments. Whereas such cross-covariances represent true statistical
covariance, they do not represent atmospheric fluxes.





## 3. Results and Discussion

### 3.1. Instrument intercomparison

Figure 2 shows the time series of O3 from all three sensors from RF03 as an example. Panel A shows the altitude of the GV, and Panel B shows the water vapor and ozone time series for the entire flight. Water vapor/ozone time synchronization was performed as close to the beginning and the end of the flight as correlation events could be visually identified; close-ups of these events are shown in Panels C and E. For EC flux legs, time synchronization was performed before and after each leg rather than for the entire flight. Before the covariance synchronization, the ozone time stamps differed from the VCSEL time stamp by up to five seconds. The beginning and end close-ups show that the synchronization procedure matches the signals to within ± 0.1 s. However, Panel D shows that the ozone signals are not synchronized with each other or to a water vapor correlation event midway through the flight. The discrepancy could be caused by a combination of different flow conditions at different altitudes or instrumental clock drift. However, because the FAIRO instruments share an inlet line that forks only in the last ~0.5 m, inlet line flow differences alone cannot explain their time offset. Inspection of the delays from each RF show that the ozone clocks drift no more than ±0.7 s (typically ≤0.5 s). For ozone instrument comparisons, data were averaged over 10 s to prevent bias from synchronization errors.

Aggregated data from RF03-07 are shown in Figure 3, Panel A. Ozone vmrs measured by each FAIRO are plotted against ozone vmrs measured by Fast O3. Linear fits of the FAIRO vmrs are also shown. The data from both instrument designs appear to be linear, with a 2% overall difference.





In Panels B-D, the absolute and relative differences between each instrument and the average of all three instruments are shown (see Fig. 3). In all cases, data are color-coded for high water

vapor (concentration $> 1.4 \times 10^{17}$ molec cm$^{-3}$, blue) and high NOx (volume mixing ratio $> 200$ pptv, red). GPS altitude, VCSEL water vapor concentration, and NOx are also shown in the background of Panels B, C, and D, respectively, for context. Neither the absolute nor the relative differences from the average exhibit systematic behavior depending on water vapor or NOx. The effect of humidity changes did not reveal any obvious explanation for O$_3$ differences when

comparing individual instruments to the instrument mean (not shown; for the effect of changing humidity when comparing FAIRO instruments, see Supplementary Figure S1). Although the persistent differences from average are accompanied by high NOx conditions between ~00:00-02:00 on 21 April 2022 UTC, high NOx conditions between ~19:30-20:00 on 20 April 2022 UTC are not accompanied by similar differences. High water vapor during a low-altitude flux leg

at 21:00 UTC is accompanied by agreement amongst all three instruments within 2%.

The agreement of the instruments was also evaluated individually for RFs 03-07. The fit results for each flight are shown in Table 1.

A subset of flux legs with low ozone variability was used to infer an upper limit of the precision of each instrument, as the contribution by additional atmospheric variability can't be fully

eliminated. The ozone time series from each flux leg is smoothed over one second and the range is calculated of the smoothed data. Variability is calculated as the relative range of ozone in that leg. Flux legs are characterized as low-variability if the relative range of the smoothed time series is less than 5% for at least two instruments. Precision is calculated as the standard deviation of the unsmoothed time series. All three instruments have comparable precision, with



Fast O3 precision at 1.4% (0.45 ppb) at 10 Hz, FAIRO 1 at 1.2% (0.36 ppb) at 12.5 Hz, and

FAIRO 2 at 1.1% (0.36 ppb) at 12.5 Hz. These precision estimates represent an upper bound as

some of the variability could be true atmospheric variability. More detail on the precision

calculations can be found in Table S2 of the supplement.

### 3.2. Eddy covariance flux

Potential temperature ($\Theta$) and equivalent potential temperature ($\Theta_e$) profiles are used to

determine whether flux legs were conducted within the MBL. Example profiles are shown in

Figure 4.

The profiles shown are taken from both descent and ascent except in the case of the RF03-C flux

legs, which were performed as the plane approached the airport for landing. The flux legs in

RF04 were conducted over the tropical Pacific Ocean, and both profiles indicate an MBL height

of ~800 m. The utility of $\Theta_e$ in determining the MBL height is evident in the RF06-A legs, which

were conducted off the coast of Alaska. The $\Theta$ profile on the descent does not unambiguously

show an MBL height, but the $\Theta_e$ profile clearly indicates an MBL height of ~200 m on both the

descent and ascent. Such a shallow MBL near the Kenai Fjords combined with the strong

temperature inversion suggests RF06-A may be subject to distinct "pools" of air; yet the $\Theta_e$

profiles suggest mixing to the surface. The MBL height in RF03-A is difficult to distinguish and

may be ~500 m. In all cases, flux legs were conducted at heights well within the MBL except in

RF03, where flux legs were conducted at 107, 476, and 889 m.

The time delay between VCSEL and the wind data is determined by calculating the water vapor

flux. VCSEL and wind speed data are well-synchronized; in most (12) cases, the water vapor

cross-covariance had a peak at a time lag of zero points; in six cases the optimal lag was -1 point





on the 12.5 Hz FAIRO time stamp (within 0.08 s). In one case (RF07-A-4) the VCSEL cross-covariance peaked at +5 points, but this is likely a spurious correlation because the VCSEL data from the previous leg was well-synchronized (zero time lag).

Because water vapor fluxes are strong and always above detection, the VCSEL/wind time offset allows us to anchor the ozone time offset and limit our search for an ozone covariance peak to ±0.7 s from the VCSEL time offset since that is the maximum observed ozone/VCSEL time delay. An ozone flux is reported for an instrument only if a cross-covariance peak is found within that window. A flux is not reported for an instrument if the covariance behavior within that

window is primarily one of sign-change, e.g. if the covariance linearly increases from negative to positive, or if there are many zero-crossings. If all three instruments show a covariance that survives this filter, then an average flux is reported for that leg. Of the 19 flux legs, 11 had fluxes that met this criterion, and are summarized in Table 2. A full version of Table 2 with meteorological conditions and other compounds of interest is included in the Supplementary

Information.

### 3.3. Comparison with literature

A previous study has compared ozone EC flux measurements from dry chemiluminescence ozone instruments over grassland (Muller et al., 2010), but to our knowledge no instrument intercomparisons have been performed on board aircraft. Aircraft measurements of ozone flux

have been reported before over land (Wolfe et al., 2018, 2015; Lenschow et al., 1980) and over the ocean during PASE (Conley et al., 2011). In the latter, ozone exchange velocities were $-0.024 \pm 0.014$ cm s$^{-1}$. Larger data sets for marine ozone flux have been produced by ship campaigns. The TexAQS cruise reported ozone exchange velocities as large as $-0.81 \pm 0.27$





cm s$^{-1}$ in coastal channels and -0.034 ± 0.003 cm s$^{-1}$ in offshore areas, and the STRATUS cruise

measured -0.009 ± 0.001 cm s$^{-1}$ over open ocean areas (Bariteau et al., 2010; Helmig et al.,

2006). All three instruments tested here can detect exchange velocities in the lower range

observed in the remote ocean.

### 3.4. Constraining the error estimate

Figure 5 shows three examples of covariance plots from flux legs that are representative of the

range of conditions observed. Ozone plots are shown on the left, and the corresponding plots for

VCSEL are shown on the right. For all cross-covariances, the Langford LOD is calculated by

using Γ = 30 s at the beginning and end of the cross-covariance plot, and the Langford 99% CL

LOD for Fast O3 is shown as a light gray shading. Panel A (RF03-C-2) is a case in which all

three ozone instruments measured an upward directed flux and VCSEL (Panel F) shows water

vapor directed downward toward the ocean; this case is described in more detail below. Panel B

(RF04-A-1) shows ozone depositing into the ocean and water vapor evaporating out of the

ocean. These two cases are examples of EC flux strong enough to be unambiguously identified

by all three ozone instruments, i.e. that each instrument's flux measurement is above the LOD as

defined by Langford et al. (2015).

Panel C (RF06-A-1) shows a case in which no ozone instrument derived flux is above the

Langford LOD. Viewed in isolation, no instrument's cross-covariance is convincing on its own.

However, a small candidate peak can be identified within the ±0.5 s interval.

The average exchange velocity measured by all three instruments in RF06-A-1 is -0.010 cm s$^{-1}$

with a standard deviation of 0.004 cm s$^{-1}$. The Langford RE$_{RMSE}$ for this leg corresponds to

0.0057-0.0074 cm s$^{-1}$ depending on the instrument and thus overstates the error and LOD. We

propose a modification of the Langford approach by restricting the interval Γ by calculating the

integral time scale. The integral time scale τ characterizes the period over which covariance

persists. We estimate τ by integrating outward from the peak until the integral crosses zero

(Lenschow et al., 2000). It is possible in certain cases for the calculation of τ to fail. This

happened for the VCSEL data shown in Panel H (RF06-A-1). In this case τ was estimated as the

width between the second zero-crossings from the peak.

We then apply the Langford RE$_{RMSE}$ calculation to intervals +Γ and -Γ which are τ in length and

are centered around relatively smooth areas of cross-correlation near the candidate peak. The

99% LOD calculated in this modified approach is shown in Figure 5 as dark gray shading. The

RE$_{RMSE}$ estimated by the modified approach corresponds to 0.0053-0.0064 cm s$^{-1}$, which is more

in line with the "true" random error among the three measurements.

### 3.5. Spatial variability of ozone and water vapor fluxes

The fluxes of ozone and water vapor were in the counterintuitive directions during the RF03-C

legs. Water vapor was carried downwards, although the ocean surface is usually a water vapor

source by evaporation. Conversely, ozone was carried upwards, even though the ocean surface is

expected to be an ozone sink. The ozone exchange velocity in this leg (RF03-C-2) was +0.134

cm s$^{-1}$ measured at an altitude of 889 m. At a lower altitude of 476 m, (RF03-C-3), the exchange

velocity was +0.097 cm s$^{-1}$. These velocities are consistent with the lower range of nocturnal

entrainment velocities (0.12 – 0.72 cm s$^{-1}$) measured during the DYCOMS-II campaign over the

Eastern Pacific Ocean (Faloona et al., 2005). However, entrainment cannot explain these fluxes

because the air above the plane was dryer, as can be seen from the profiles in Figure 6.





A plausible source of water vapor above the plane is evaporating cloud droplets; indeed, footage

from this RF shows the plane flying below numerous low clouds. RF03-C took place near solar

noon, so the $NO_2$ photolysis frequency ($J_{NO2}$) measured by the HIAPER Airborne Radiation

Package (HARP) actinic flux instrument can be used as a proxy for overhead cloudiness (see top

panel of Figure 6). A period of high ozone and water vapor flux between 22:41 and 22:43 UTC is

accompanied by oscillations in $J_{NO2}$ that indicate heavy cloud cover. Stills from the flight movies

are shown in Figure 7.

In the left panel of Figure 7, the forward- and side- looking cameras show light high-altitude

clouds corresponding to a time (22:39:59 UTC) of low $J_{NO2}$ variability and fluxes near zero for

both ozone and water vapor. The right panels show the same views with heavy low clouds

visible, corresponding to a time (22:41:57 UTC) when the $J_{NO2}$ variability is high, water vapor

flux is toward the ocean, and ozone flux is toward the clouds above.

We hypothesize that evaporating clouds provide a source of water vapor while simultaneously

providing a sink of ozone. Previously, it was proposed that an increase in aqueous phase

chemistry in cloud droplets would decrease ozone production in high-NOx environments and

enhance ozone destruction in low-NOx environments (Lelieveld and Crutzen, 1990).

Computational simulations suggest that ozone could be stabilized within the air-water interface

(within the first 4 Å), and that modification of the ozone UV-vis absorption cross section and

activation of photolytic pathways at the interface can increase the ozone photolysis rate constant

by more than a factor of 20 (Anglada et al., 2014). The observations from the RF03-C legs may

represent the first field evidence of these proposed processes. Critically, the RF03-C-1 flux leg

performed at 107 m immediately prior to the RF03-C-2 887 m leg found fluxes below detection





for all three ozone instruments. Thus, if cloud effects are operative, they may well be invisible to

surface-based platforms such as ships.

Compared to shipborne measurements taken over the course of days or weeks, the flux legs here are necessarily shorter, with the longest leg being ten minutes and the legs being only ~5 minutes long on average. To assess the consistency between sensors on shorter time scales, the ozone EC fluxes were also calculated in 75-second long quarters for the flux example RF06-A-1 shown in

Figure 7.

The ozone flux observed in this leg is carried in the first, third, and last quarters, with flux in the second quarter below detection. However, the water vapor flux is above detection in all segments and exhibits different trends from the ozone flux. Since the water vapor and ozone are both carried by the same eddies, the difference in behavior cannot be attributed to meteorology.

Rather, the ozone flux variability must reflect true heterogeneity in the ocean and/or atmospheric chemical states. Assuming chemical measurements are available on similar time scales, the ozone flux can help characterize atmospheric chemistry on ~10 km spatial scales. For measuring average fluxes, we recommend flying multiple flux legs over regions of interest for better statistics as ozone fluxes are often near the LOD.

## 4. Conclusions and Outlook

In the aggregate, Fast O3 and FAIRO instruments operate at comparable frequencies (10 vs 12.5 Hz data rate; 3 Hz practical resolution estimated from the mixing time of zero-air puffs at the Fast O3 inlet), are accurate within 2%, and have similar LOD at their typical sampling rates (1.5 ppbv). Large excursions in measured ozone vmrs (of up to 30%, or 5 ppbv difference) are

sometimes observed in the ratio of high-rate data between the instruments, but the excursions





show no systematic behavior with respect to ozone concentration, water vapor or NOx. These differences did not occur during the flux legs. From an operational standpoint, the FAIRO design is advantageous, because the instrument and pump fit into a single 19" rack and requires no hazardous NO gas.

Simultaneous, high-frequency $H_2O$ measurements in the free stream are essential for synchronizing the $O_3$ sensors and wind measurements, and provide context to the interpretation of $O_3$ EC fluxes. Inlet line delays, clock drifts, and small inaccuracies in clock synchronizations lead to time offsets that are difficult to characterize with certainty. Correlation events between water vapor and ozone present direct means for clock synchronization. In principle, an ozone

time lag could be prescribed by matching the ozone time stamp to the water vapor time stamp and searching for time lag at which water vapor flux peaks since the water vapor flux is always above detection. In practice, clock drifts still necessitate a search for a cross-covariance peak in the ozone flux, albeit in a constrained time window.

The availability of three ozone instruments during TI³GER allowed for the estimation of the

"true" LOD of the ozone flux ($LOD_{ECflux}$) using the standard deviation of the EC fluxes measured by each instrument. We use this information to provide a modified procedure to estimate error and $LOD_{ECflux}$: the $RE_{RMSE}$ formula (eq. 7) (Langford et al. 2015) is combined with the concept of "integral time scale" (Lenschow et al., 2000). We find that the "true" $LOD_{ECflux}$ (defined as the 95% CI on the mean EC flux) is overestimated by the EC flux

uncertainty on an individual sensor when $\Gamma$ is a large time window (30 sec, as used in Lenshow et al.). Estimating the $RE_{RMSE}$ over a smaller time window shrinks the $RM_{RMSE}$, and brings the EC flux uncertainty closer to the "true" error inferred from the EC flux standard deviation of



three separate sensors, without underestimating the EC flux error. We find that the integral time

scale $\Gamma$ suitable to estimate error is usually a few seconds, and define it here as $\Gamma$ found by

integrating outward from a candidate covariance peak until the first zero-crossing of the

covariance integral. Typical LODs for $O_3$ exchange velocities are 30-50% lower with shorter $\Gamma$,

with typical LOD ~ 0.005 cm s$^{-1}$, limited by spurious covariance peaks that are clearly non-

physical as they exceed the believable bounds of instrument synchronization.

Ozone EC fluxes measured from aircraft in the remote MBL can exhibit significant time

variability on the order of minutes (6-10 km). A similar variability is not seen in the $H_2O$ EC

fluxes. While the $H_2O$ EC fluxes are spatially more homogeneous, and de-facto constant (within

25%), a variability in the $O_3$ EC fluxes of larger than 600% is observed and highly significant

(above 6-$\sigma$ to below detection) on spatial scales of 20 km. This variability is seen consistently by

all three sensors over the open ocean environments probed here. Cloud cover can reverse the

direction of the $O_3$ and $H_2O$ fluxes, indicating a source of water vapor and a sink for $O_3$ above

the aircraft, consistent with webcam images of clouds. The drivers of the horizontal variability in

$O_3$ EC fluxes directed into the ocean on fine spatial scales is currently not well understood, but

could relate to changes in overhead cloud cover, as well as possibly variability in ocean and

atmospheric states. Future studies are needed, and would benefit from repeat legs, and

measurements of ocean state variables. can reverse the direction of the $O_3$ and $H_2O$ fluxes,

indicating a source of water vapor and a sink for $O_3$ above the aircraft, consistent with webcam

images of clouds. The drivers of the horizontal variability in $O_3$ EC fluxes directed into the ocean

on fine spatial scales is currently not well understood, but could relate to changes in overhead

cloud cover, as well as possibly variability in ocean and atmospheric states. Future studies are

needed, and would benefit from repeat legs, and measurements of ocean state variables.



**Code Availability**

The MATLAB flux toolbox is available at: https://github.com/AirChem/FluxToolbox.

**Data Availability**

All data used in this paper can be found on the TI3GER field catalog, which is available at the

following URL: https://www.eol.ucar.edu/field_projects/ti3ger.

**Author Contributions**

RV designed the TI$^3$GER project, and as mission scientist planned and led research flights. RC

performed data analysis of the EC fluxes and instrument intercomparison, assisted with

instrument calibrations and uninstallation, and led the manuscript preparation. FO and AZ

calibrated and deployed the FAIRO instruments and provided the FAIRO data. AF and TC

calibrated and deployed the Fast O3 instrument and provided Fast O3 data. AR and CW

calibrated the wind measurements and provided GV data. RC and RV wrote the manuscript, with

contributions from all co-authors.

**Competing Interests**

At least one of the (co-)authors is a member of the editorial board of Atmospheric Measurement

Techniques.



**Acknowledgements**

Financial support for TI³GER from US National Science Foundation award AGS-2027252 (PI: R. Volkamer) is gratefully acknowledged. RC and RV thank Glenn Wolfe, Erin Delaria and

Reem Hannun, and Dongwook Kim for helpful discussions. TI³GER was supported by the National Center for Atmospheric Research, which is a major facility sponsored by the NSF under Cooperative Agreement no. 1852977. The data were collected using NSF's Lower Atmosphere Observing Facilities, which are managed and operated by NCAR's Earth Observing Laboratory. The GV aircraft was operated by the National Center for Atmospheric Research (NCAR) Earth

Observing Laboratory's (EOL) Research Aviation Facility (RAF). The NCAR ozone measurements were funded by NSF Lower Atmosphere Observing Facilities and NSF NCAR/Facilities programs.

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






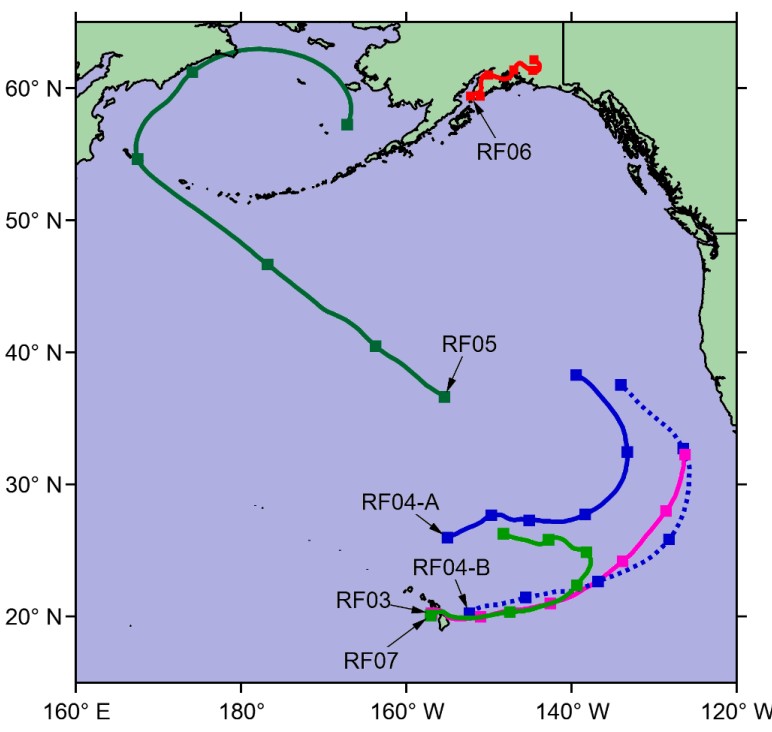

Figure 1. Map of flux legs and back-trajectories during TI3GER. Square markers indicate 24-hour periods, and the arrows mark the location of the flux legs.



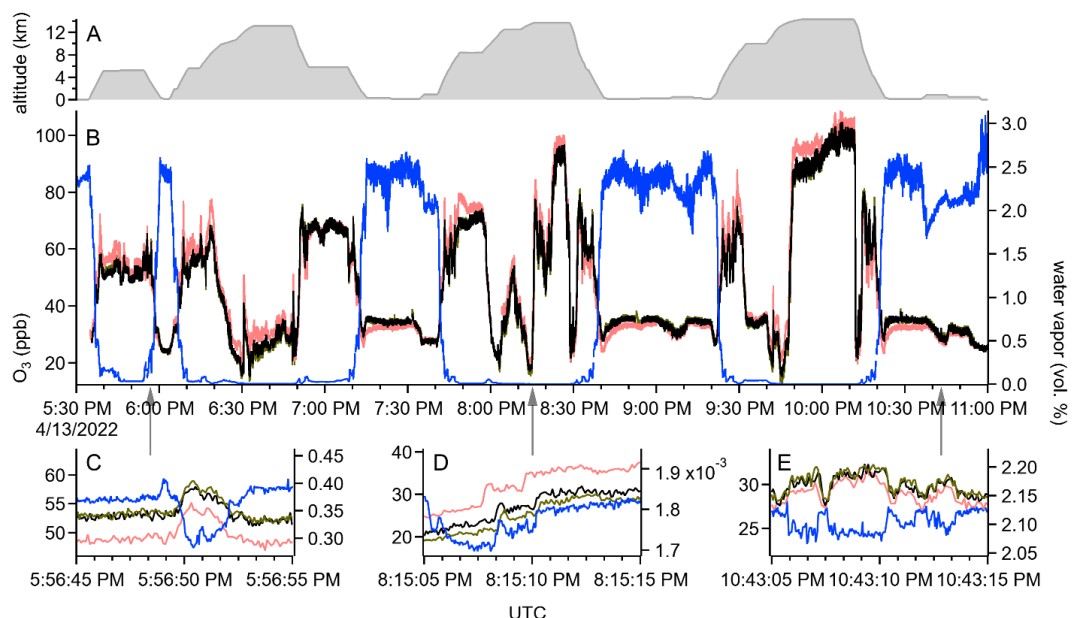

Figure 2. Time stamp synchronization based on the $H_2O$ and $O_3$ time series. VCSEL data are shown in blue, Fast O3 data in salmon, FAIRO 1 in black, and FAIRO 2 in dark olive. All traces are shown at the native instrument resolution (25 Hz for VCSEL, 12.5 Hz for the FAIROs, and 10 Hz for Fast O3). A: altitude time series. B: time series for the entire flight. C-E: Zooms to cross-covariance events with gray arrows pointing to exact times.





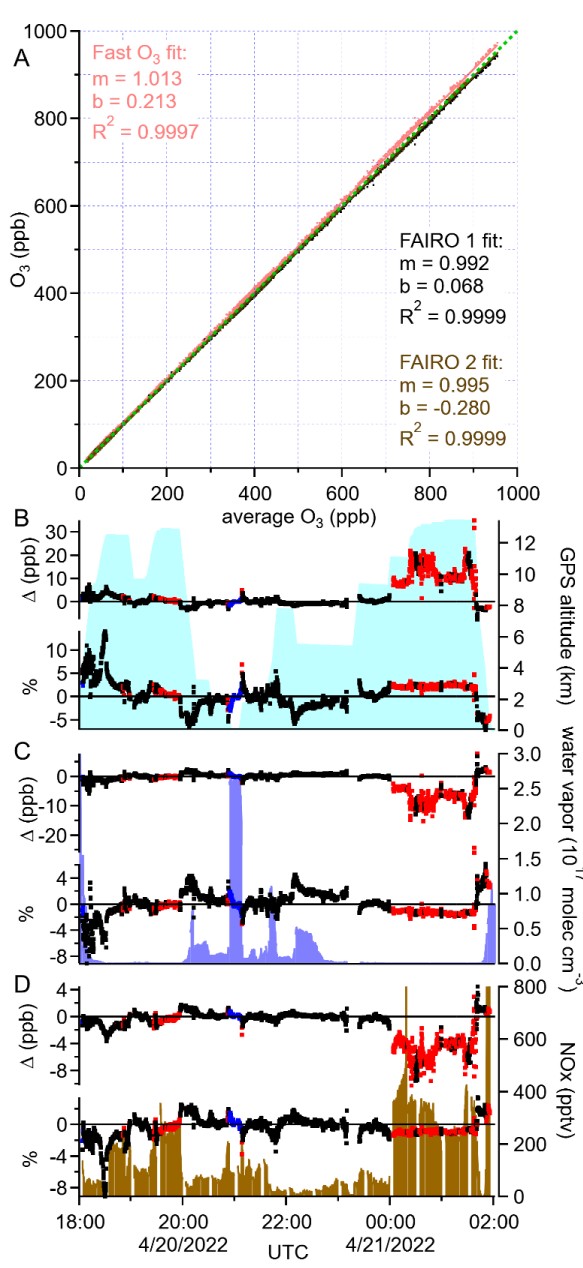

Figure 3. A: Aggregated data from RF03-07 with fits relative to global average; one-to-one line in green. Absolute
and relative differences from average during RF05 for Fast O3 in Panel B; FAIRO 1 in Panel C; and FAIRO 2 in
Panel D. Background shading for GPS altitude in Panel B, VCSEL in Panel C, and NOx in Panel D.

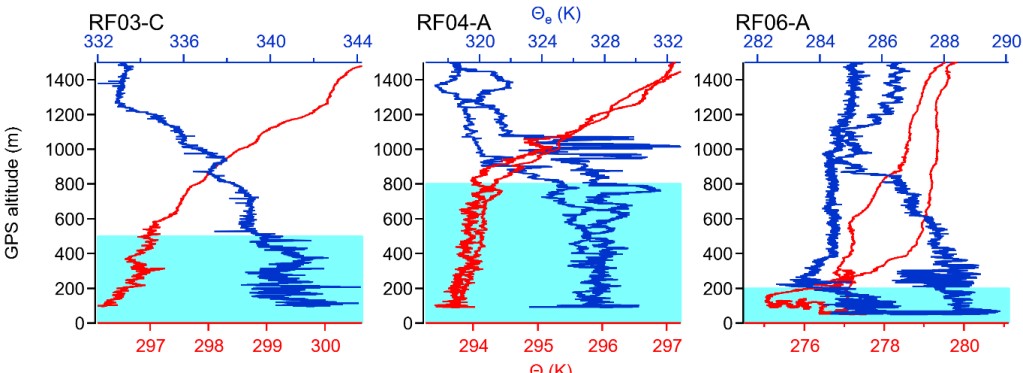

Figure 4. Representative potential temperature and equivalent potential temperature profiles for determining MBL height in three different flux legs. The MBL is shown as light blue shading.

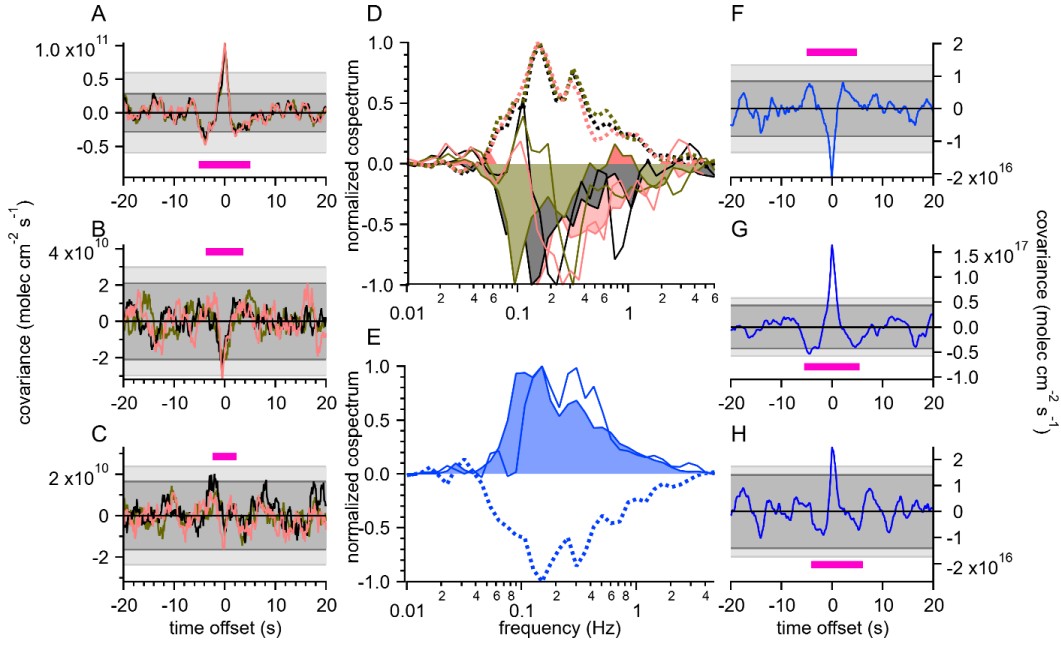


Figure 5. Cross-covariance plots for RF03-C-2 (A), RF04-A-1 (B), and RF06-A-1 (C), and their respective water vapor fluxes (F-H). Normalized cospectra are shown in D and E. For ozone data, Fast O3 is shown in salmon, FAIRO 1 in black, and FAIRO 2 in olive. In Panels D and E, RF03 is shown as a dotted line, RF04 as a shade to zero, and RF06 as a solid line. Integral time scales are shown as fuchsia bars.

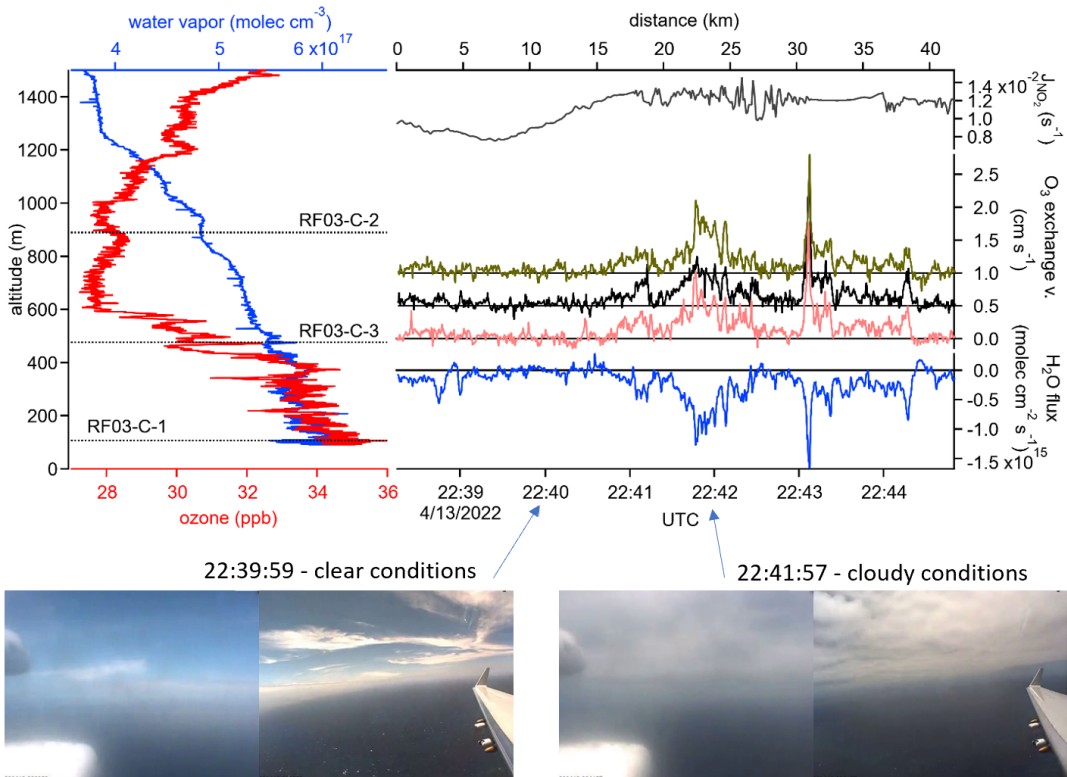


Figure 6. Ozone and water vapor vertical profiles and time series for EC fluxes from the RF03-C leg. Ozone profile is the average of all three instruments. Dashed lines indicate flight altitudes. Right: $J_{NO2}$ in gray, fluxes from Fast O3 (salmon), FAIRO 1 (black), FAIRO 2 (olive), and VCSEL (blue). Vertical offsets of 0.5 cm s$^{-1}$ and 1 cm s$^{-1}$ have been added to FAIRO 1 and 2 to better illustrate the close agreement between the three $O_3$ instruments. Images of
the webcams from RF03 flight movies illustrate cloud cover conditions.



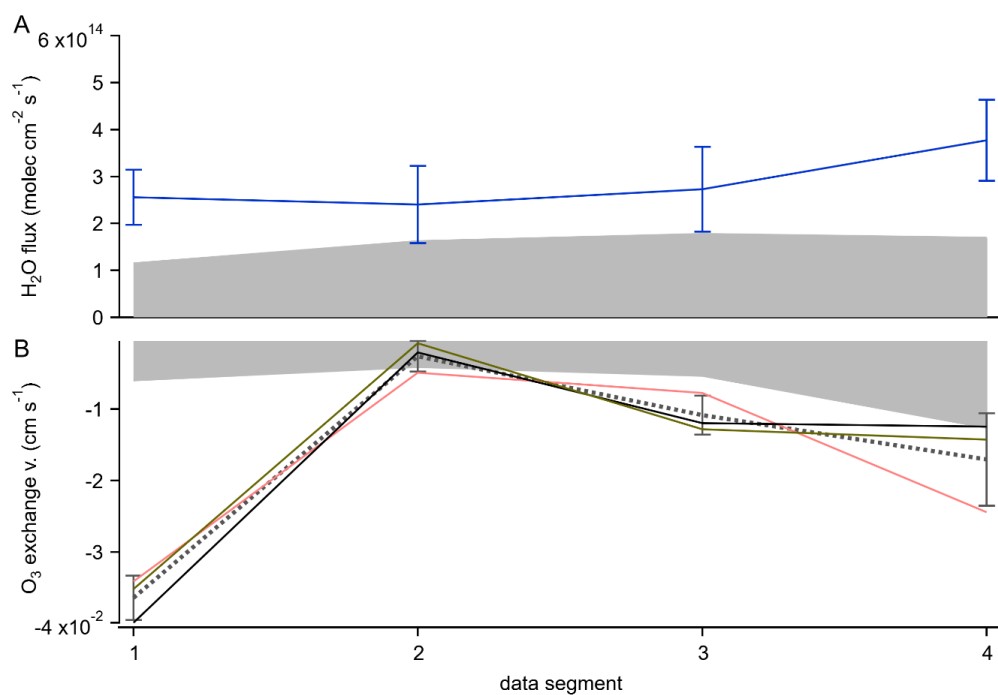

Figure 7. Segments of RF06-A-1. Panel A: VCSEL is shown in blue. Error bars represent modified Langford RE$_{RMSE}$. Panel B: FO3 in salmon, FAIRO 1 in black, and FAIRO 2 in dark olive. The average is shown as the dotted line. Error bars represent standard deviation. In both panels, the 95% LOD is shaded. Each data segment is 75 seconds long.

| flight | average (ppb) | max (ppb) | FO3 slope | FO3 offset (ppb) | F1 slope | F1 offset (ppb) | F2 slope | F2 offset (ppb) |
|---|---|---|---|---|---|---|---|---|
| RF03 | 47.4 | 103 | 1.028 | 0.44 | 0.982 | 0.02 | 0.990 | -0.46 |
| RF04 | 76.5 | 409 | 1.029 | -0.03 | 0.981 | 0.24 | 0.989 | -0.21 |
| RF05 | 172.8 | 955 | 1.024 | -1.40 | 0.985 | 1.10 | 0.991 | 0.34 |
| RF06 | 223.0 | 887 | 1.000 | 0.15 | 0.998 | 0.15 | 0.998 | -0.30 |
| RF07 | 78.6 | 177 | 1.014 | 0.08 | 0.992 | -0.04 | 0.994 | -0.04 |

Table 1. Linear fit parameters of individual instruments to average.





| Leg Code | Date UTC | Start UTC | End UTC | location | alt (m) | exchange velocity (cm s⁻¹) | | | | |
|---|---|---|---|---|---|---|---|---|---|---|
| | | | | | | Fast O3 | FAIRO 1 | FAIRO 2 | average | st. dev. |
| RF03-A-1 | 4/13/2022 | 19:16:43 | 19:20:13 | off SW coast of HI | 312 | +0.037 | +0.020 | +0.015 | +0.024 | 0.012 |
| RF03-B-1 | 4/13/2022 | 20:45:33 | 20:52:08 | off SW coast of HI | 101 | +0.033 | +0.014 | +0.003 | +0.017 | 0.015 |
| RF03-C-2 | 4/13/2022 | 22:38:15 | 22:44:51 | off SW coast of HI | 889 | +0.131 | +0.135 | +0.136 | +0.134 | 0.003 |
| RF03-C-3 | 4/13/2022 | 22:46:02 | 22:54:15 | off SW coast of HI | 476 | +0.099 | +0.093 | +0.100 | +0.097 | 0.004 |
| RF04-A-1 | 4/15/2022 | 21:16:20 | 21:21:46 | North of HI | 93 | -0.042 | -0.037 | -0.030 | -0.036 | 0.006 |
| RF04-A-2 | 4/15/2022 | 21:22:12 | 21:25:50 | North of HI | 101 | -0.030 | -0.014 | -0.017 | -0.021 | 0.008 |
| RF06-A-1 | 4/21/2022 | 19:39:47 | 19:44:44 | off AK coast | 58 | -0.015 | -0.008 | -0.009 | -0.010 | 0.004 |
| RF06-B-1 | 4/22/2022 | 02:28:59 | 02:32:01 | halfway between AK and HI | 116 | +0.022 | +0.024 | +0.024 | +0.023 | 0.001 |
| RF07-A-1 | 4/23/2022 | 21:43:30 | 21:49:20 | west of HI airport | 116 | -0.017 | -0.013 | -0.014 | -0.015 | 0.002 |
| RF07-A-3 | 4/23/2022 | 22:04:07 | 22:06:45 | west of HI airport | 778 | -0.012 | -0.015 | -0.015 | -0.014 | 0.002 |
| RF07-A-4 | 4/23/2022 | 22:11:24 | 22:16:08 | west of HI airport | 472 | -0.035 | -0.029 | -0.036 | -0.033 | 0.004 |

Table 2. Summary of ozone EC flux results.