# Peer review of "Intercomparison of Fast airborne Ozone Instruments to measure Eddy Covariance Fluxes: Spatial variability in deposition at the ocean surface and evidence for cloud processing"

_Atmospheric Measurement Techniques, 2023_

## Author Comment (AC1)

**REVIEW #1**

**Overview:**

Chiu et al. present a comparison of airborne measurements of ozone mixing ratios and vertical fluxes in the marine boundary layer from three different fast ozone sensors recently deployed on a GV aircraft. Measurements from the three instruments were shown to compare well and overall flux uncertainty was assessed from mean and deviation of the three flux measurements which is a useful addition to understanding of ozone flux uncertainties. The authors also present a modified flux limit-of-detection method building on the cross-covariance method of Langford et al. (2015) but with refinement of the time scales which is a nice addition. Finally the authors present some case studies to demonstrate the utility of the observed ozone exchange velocities which potentially imply reactions on cloud droplets as a sink of ozone and spatial variability in ozone deposition to the ozone surface on small spatial scales. My two broad comments are 1.) that while the measurements seem to be of a high quality, some relevant details of the flight campaign and data processing are lacking in places (see specific comments below), and 2.) that the analysis of the case studies are underdeveloped and conclusions of cloud loss and spatial heterogeneity need more support if they are to be included. I appreciate that these case study results are not the primary focus of this methods paper, but the authors are still making conclusions about ozone loss to clouds that I believe need more support. Some specific comments on this are included in the major comments below. Overall the comparison of the airborne ozone measurements is novel and valuable and is suitable for AMT with some revision. The interpretation of the case studies seem reasonable but require additional information in order to support the conclusions of the authors. If the above points are addressed then this work will likely be a useful contribution to AMT.

**Major Comments:**

Q1: Line 96: Some additional discussion of the EC flux leg design is needed in the main text, not just table S1. What were the lengths of these flux legs? What is the airspeed of the aircraft? Were stacked legs flown at multiple altitudes in the same location? Were vertical profiles out of the boundary layer performed to constrain the potential entrainment term?

A1: Regarding the flights, the following text has been added to the revised manuscript:

Flux legs were typically 5-10 minutes long. At airspeeds of $\sim$110 m s$^{-1}$, flux legs covered 30-70 km. A typical flight module consisted of three legs flown in a stacked manner (RF03-B, RF03-C, and RF07-A). However, in the case of RF03-B, fluxes were below detection. Hence, other flight legs were opportunistically used for flux calculations on level legs in the marine boundary layer (MBL). Dedicated flux segments were accompanied by profile descents and ascents.
A discussion of entrainment has been added and is addressed elsewhere (see our response to the comment on section 3.5).

Q2: Relatedly, basic details of the eddy covariance data processing and quality control are missing. For example, was there a stationarity criteria applied to the flux data, what is the impact of high pass attenuation for the closed path sensors and was a correction applied. What are the random and systematic turbulent sampling errors (Lenschow et al. 1994, https://doi.org/10.1175/1520-0426(1994)011%3C0661:HLILEW%3E2.0.CO;2) ? These types of details are needed for a proper evaluation of the methods and results presented.

A2: Regarding stationarity, we have added the following text to the revised manuscript:

Stationarity is not required for wavelet analysis (WA) because WA decomposes the total flux into component fluxes at different frequencies.

Regarding the frequency response of closed path sensors, the following text along with two new SI Figures have been added to the revised manuscript:

The cospectra in Fig. 5 peak from 0.1-0.2 Hz, indicating that the bulk of the fluxes occur at 5-10 s time scales. These timescales are typical for fluxes in the MBL, and an order of magnitude larger than the mixing time for the Fast $O_3$ instrument, which for background chracterization purposes had zero-air injected in from the aircraft inlet. The e-fold rise time was <0.5 s, fast enough not to introduce bias to the flux measurements (see Figure S3 in the SI). Indeed, the cumulative frequency graph (ogive) shows that in the case of RF03-C-2, less than 10% of the total flux is carried on <1 s timescales. Ogives are shown in Figure S4. The residence time in the fast ozone instrument detection volume implied a maximum frequency response of 9 Hz. However, high pass attenuation in the inlet manifold limited the frequency response of the fast O3 instrument to 3 Hz (Lenschow & Raupach, JGR 96(D8), 15,259-15,268, 1991). The FAIRO instruments were not equipped with zero-air injection at the inlet. However, the residence time in the FAIRO flow is shorter than that in Fast O3. A calculation of  FAIRO inlet manifold attenuation of high frequency signals above 20 Hz, and therefore was not the limiting factor in the FAIRO instrument frequency response, Thus the FAIRO was more sensitive to high-frequency fluxes than the Fast O3 instrument.

Figures S3 and S4, which have been added to the revised SI, appear below:

[Figure]

Figure S3: Fast O3 photon counts during valve-switching from zero air back to sampling outside air from the inlet. Data points are collected at 10 Hz.

[Figure]

Figure S4: Cumulative frequency graphs (ogives) for RF03-C-2 (black) and RF04-A-1 (red) of the ozone fluxes from the Fast O3 instrument. Solid lines: detrended to 10 s; dotted lines: no detrending. Detrending eliminates fluxes over >10s time frames. Fluxes at frequencies >1 Hz account for ~10% of total flux.

Q3: Line 255: The large time desynchronization mid-flight mentioned around line 255 even after applying the manual synchronization method is concerning. Inlet flow would have to vary strongly as a function with altitude or clock drift would seemingly have to be non-linear to describe this behavior. Were either of the inlets for the ozone instruments pressure and or mass flow controlled? What implications are there for systematic flux uncertainty if flow rate changed significantly as a function of altitude?

A3: The text has been revised to clarify what appears to be a misunderstanding here. The 5 s time offset is from before manual synchronization of computer clocks; this is an artifact which occurred on an

individual flight when the instrument computer's clock was not being synchronized with the time server. We mention this because the ability of the method for manual time synchronization based on O3 and H2O is found to be capable of capturing even such large artificial delays; this should thus be reassuring rather than worrisome. After time synchronization, delays typically less than 0.5 s, consistent with internal flow differences between closed path sensors. We have clarified this in the revised manuscript:

Because the Fast O3 instrument computer was not synchronized with the time server, there was an artificial delay of 5 s between it and VCSEL. After the time synchronization procedure, even artificial clock delays are resolved to within ±0.1 s.

The flow through both instruments is described in the manuscript:

"The total mass flow in the inlet was 2370 sccm." for Fast O3 and "Outside air was pulled at 11 vol.-L min$^{-1}$ at ambient pressure" for FAIRO.

The implications for systematic error in the fluxes is addressed in this new text in the revised manuscript:

Whereas the Fast $O_3$ instrument used constant mass flow at constant pressure, the FAIRO instruments used constant volume flow at ambient pressure. In principle, the flow rates in the two instrument designs could differ between the high altitude/low pressure legs typically used for time synchronization and the low altitude/high pressure legs used for flux measurements. Although the different flow rates can create time lag between wind and ozone data, no systematic error is introduced to the ozone flux because we empirically determine the time offset, and do not prescribe a constant offset in the MATLAB flux toolkit. Rather, the time synchronization is used in conjunction with water vapor fluxes calculated from VCSEL data to find the true ozone time offset.

Q4: Figure 5. Some additional analysis of the spectral response would be appreciated in the SI. Generally it would be useful to validate that there is no significant high-frequency attenuation for the ozone instruments and comment on any attenuation corrections applied.

A4: We address this along with the previous comment regarding volumetric flow rates and stationarity with the addition of the following text (see A2).

Q5: Section 3.5: My primary comment on the manuscript relates to this section as a whole. There are limited conclusions that you can draw from an eddy covariance flux measurement at a specific altitude in isolation. Interpreting the measured flux values to infer source or sink terms requires some knowledge of the vertical profile of (here) ozone mixing ratios and fluxes. The magnitude of the surface deposition and the entrainment flux at the top of the boundary layer influence what the measured flux magnitude (and direction) will be at some altitude in the boundary layer in addition to chemical source and sink terms at the measurement altitude. The O3 vertical profile shown in Figure 6 shows lower O3 mixing ratios aloft then a weak turnover at ~1200 m, however it is not clear what the boundary layer top is and what the profile is above that. While the measured ozone flux could be indicative of loss on evaporating cloud droplets, there may be contributions from entrainment of ozone depleted air from the free troposphere into the boundary layer. At a minimum some marking of the boundary layer height, ozone mixing ratios above the boundary layer, and the cloud depth would be quite useful on this figure. More generally I encourage you to present these results as vertical flux divergence figures ( see Wolfe et. al. (2015) () and Conley et. al. (2011) for examples). Some estimate of entrainment velocities in the clear sky case could also be useful both as a valuable result in their own

right, but also to aid in interpretation of the cloud loss argument (see Faloona et al., https://doi.org/10.1175/JAS3541.1)

A5: We agree, and are providing more context to our observations based on this feedback:

[revised manuscript text omitted]

Q6: Similarly for Fig 7 and the related discussion, I agree that the observed variability in measured O3 exchange velocity is not due to turbulence and reflects some variability in the oceanic or atmospheric chemical state. However you do have at least some additional information that can be used to constrain this observed behavior. Are O3 mixing ratios changing significantly across the flux segments? How do the vertical profiles and the beginning and end of this segment compare?

A6: Because the RF03-C flux legs were performed on approach to the airport for landing, only a descent profile is available. Ozone profiles have also been added to Fig. 4, included below:

[Figure]

Figure 2. A-C: Profiles of ozone during RF03-C, RF04-A, and RF06-A, respectively. D-F: Corresponding potential temperature and equivalent potential temperature profiles for RF03-C, RF04-A, and RF06-A, respectively. MBL height is shown as light blue shading. Arrows indicate profile ascents and descents.

Overall, I don't think the interpretations suggested by the authors are unreasonable, however they should be presented in a more comprehensive way with more relevant constraints.

**Minor Comments:**

Q7: Line 36: The implications of the observed high variability of v(O3) should be made more explicit in the abstract.

A7: The revised abstract now reads:

Additionally, we present two case studies. In one, the direction of ozone and water vapor fluxes were reversed (vO3 = +0.134 ± 0.005 cm s-1), suggesting that overhead evaporating clouds could be a strong ozone sink. Further work is needed to better understand the role of clouds as a possibly widespread sink of ozone in the remote marine boundary layer. In the second case study, ozone fluxes vO3 are negative (varying by a factor of 6-10 from -0.036 ± 0.006 to -0.003 ± 0.004 cm s-1), while the water vapor

fluxes are consistently positive due to evaporation from the ocean surface and spatially homogeneous. This case study demonstrates that the processes governing ozone and water vapor fluxes can become decoupled, and illustrates the need to elucidate possible drivers (physical, chemical, or biological) of the variability in ozone exchange velocities on fine spatial scales (~20 km) over remote oceans.

Q8: Line 61: Much is made of the rarity of comparison studies of ozone flux but there is no motivation for specific knowledge gaps that such comparisons can provide. To my knowledge there has not been much suggestion in the literature that instrument uncertainty is a major driver of overall uncertainty in parameterizing ozone deposition. Some synthesis of the literature and implications from previous comparison studies would be welcome here to motivate this study.

A8: We appreciate this comment, and agree with the reviewer.

We also note, that for example reviewer #2 suggests instrument uncertainties are caused by water sensitivities as major uncertainties in ozone fluxes. This is also a view in the community. Our responses below makes it clear that these water sensitivities are indeed small. We thus agree with reviewer #1. However, there is no previous intercomparison of different instruments that rely on different measurement concepts and respond differently to water sensitivities on research aircraft. We have clarified this in the revised manuscript, along with other reasons that motivate this work.

Q9: Line 70: Hannun et al. (https://doi.org/10.5194/amt-13-6877-2020) presented airborne EC flux results over the ocean and a comparison of mixing ratio measurements from a new broadband cavity-enhanced UV absorption instrument with an NO chemiluminescence instrument on the NASA DC-8 aircraft during the FIREX-AQ campaign.

A9: Thank you for this helpful comment. We have made significant revisions to that introductory paragraph:

Whereas EC flux measurements of ozone are numerous, comparison studies are fewer. Ozone fluxes from EC methods have been compared to those from gradient measurements (Muller et al., 2009; Zhu et al., 2020; Loubet et al., 2013) and dynamic chamber methods (Plake et al., 2015). Over grassland, Plake et al., (2015) report that dynamic chamber methods agree "well" with EC flux methods (within 11-26%). Over maize fields, Zhu et al., (2020) describe the discrepancy between EC flux and gradient methods as "not very good," with gradient methods measuring ozone fluxes 11.7 – 45.6% higher than those measured by EC flux methods. Furthermore, comparisons of co-located EC flux measurements are uncommon, and complicated due to vertical gradients in the measured fluxes that may explain differences of 10% between measurements on towers (measured by chemiluminescence) and aircraft (measured by a TECO-49) (Massman et al., 1995). To our knowledge, the only aircraft instrument intercomparison for ozone EC flux was performed by Muller et al., (2010), who compared two identical dry chemiluminescence instrumental clones over grassland, and found differences up to 12% due to differing sensitivities of chemiluminescent discs. Furthermore, a water sensitivity for chemiluminescent measurement techniques (Ridley et al., 1992) has been suggested to propagate onto EC ozone flux measurements (Boylan et al., 2014), and methods for water correction differ between different methods for measuring ozone. More commonly, a fast ozone instrument is compared to other ozone instruments only in terms of concentrations (Conley et al., 2011; Hannun et al., 2020). There is currently no intercomparison of different fast ozone instruments that rely on different measurement concepts and respond differently to water sensitivities on research aircraft. Furthermore, the error analysis to estimate EC flux uncertainties is not well developed and is not always treated consistently. This leaves room for instrument and method uncertainty as drivers for overall uncertainty in parameterizing ozone exchange velocities and deposition. Here we eliminate spatial gradients as a source of uncertainty in ozone EC flux intercomparisons by deploying three ozone instruments of two different designs on research aircraft in remote marine air. We further use the agreement found among

the three sensors to evaluate and refine the EC flux error analysis and define better criteria of use to estimate detection limits.

Q10: Line 109: An estimate of the total volumetric inlet residence time would be useful. Same for the FAIRO instruments.

A10: The following text has been added to the revised manuscript. For Fast O3: Total residence time is ~2.5 s. For FAIRO: Residence time in the line is approximately 0.3 s.

Q11: Line 170: How many of these anticorrelation synchronization anchors do you generally get in a flight? Do you use events at all altitudes or only ones at low altitudes relevant for the flux sampling conditions.

A11: The following text has been added to the revised manuscript:

Anticorrelation events are not uncommon. For instrument intercomparison, anticorrelation events from the start and end of the entire flight are used to synchronize data; averaging the synchronized data over 10 s is sufficient to resolve any residual (<100 ms) synchronization uncertainty. For flux sampling, anticorrelation events were found before and after each flux leg.

Q12: Figure 5: What is driving the shift in the cospectra toward higher frequencies on RF6 and the rapid fall off in spectral power at frequencies below 0.1 Hz? Could be due to the detrending method or due to real atmospheric turbulent structure. Analysis of the vertical wind power spectra would help clarify.

A12: The following text has been added to the revised manuscript:

Detrending the data at 10 s removes spectral power and frequencies below 0.1 Hz.

Please also see A2, where ogives have been added to the SI.

Q13: Relatedly at Line 226, it is not clear that a fixed detrending window of 10 s is appropriate when boundary layer height and cloudiness and thus convective strength are varying between flux legs. Some additional justification would be useful.

A13: The following text has been added to the revised manuscript:

For all flux legs, various detrending times were tested to see whether visually identifiable structures could be observed in the cross-covariance. A uniform 10 s detrending time was found to remove systematic structures from all flux legs. To minimize the number of subjective inputs, we did not attempt to customize the detrending time for each flux leg. Because detrending accounts for meteorological conditions rather than instrument response, meaningful intercomparisons could be performed using uniform conditions and consistent detrending times.

Q14: Line 318: Filtering out low signal to noise flux measurements does not seem totally appropriate. These measurements provide real information on the flux magnitude (e.g. that they are below the LOD). Excluding those values from an average of the ozone flux would artificially bias the magnitude low, much like excluding below LOD gas phase mixing ratio measurements would add a high bias to the mean. This is of minor importance in this paper since flux results are mostly considered in the

context of case studies from single flux legs, but I want to raise this as a general point in the data interpretation.

A14: We do not report a global average of fluxes due to the technical nature of this campaign. We only summarize cases where fluxes are above detection.

Q15: Line 363: How are the "relatively smooth areas of cross-correlation near the candidate peak" identified?

A15: The following text has been added to the revised manuscript:

Identifying "smooth" areas was necessarily subjective as the cross-correlation behavior is unique to each leg.

**Technical Comments:**

Q16: Line 143: Give inlet diameter in cm or mm instead of inches.

A16: The revised manuscript reads:

The FAIROs sampled from a separate HIMIL (aft-facing inlet line) through a PFA line with a length of 4.3 m and a 0.42 cm (1/6 in.) inner diameter.

**Citation**: https://doi.org/10.5194/amt-2023-198-RC1

---

## Author Comment (AC2)

**REVIEW #2**

**This manuscript is unique and very much appreciated reporting on the comparison of three aircraft ozone eddy covariance flux measurements operated in parallel. As the authors correctly point out, few flux intercomparisons have previously been reported.**

While overall this paper is well written, unfortunately, in the opinion of this reviewer, it neglects a number of important issues that factor into this monitoring and research.

Q1: Clifton et al. (2019) presented a comprehensive overview of ozone deposition monitoring and ozone flux research. Unfortunately, this paper and findings are not recognized by Chiu et al.

A1: The paper to which the reviewer refers is Clifton et al. (2020), which was already cited in the manuscript. Clifton et al. (2019) is a study on the effects of humidity on the deposition of ozone onto stomata, which is not relevant to the work under review.

Q2: It has been known for a long time that water vapor can be a severe interference in various monitoring methods for the determination of ozone. The authors pay some credit to these effects, however, do not fully recognize the severity that quenching of the fast ozone signal from fast fluctuations of water vapor can have in the determination of eddy covariance ozone flux determination.

Willson and Birks (2006) pointed out that interferences in ozone measurement can be particularly high during the fast water vapor changes that can be experienced during aircraft sampling from elevation changes and when flying through clouds for UV absorption instruments. Their recommendation was to selectively remove water vapor from the sampling stream with a Nafion dryer.

R2: We appreciate the reviewer's focus on water vapor interference, but we disagree about the overall importance. The original paper had already actively dealt with the water interference, as we elaborate below.

We also have a different perspective on the relevance of Wilson and Birks (2006) to this study. First, Wilson and Birks (2006) consider interference in UV photometers by water vapor adsorption onto UV optics. This is not a relevant effect in the coumarin chemiluminescence channel of the FAIRO instrument, which the current EC flux comparison is based on. To the extent that the Fast O3 instrument may be affected by water vapor, such interference should already be accounted for by the Ridley et al. water vapor correction, which is empirically derived. The agreement of the FAIRO instruments (to which the Wilson and Birks water vapor effect is not relevant) with the Fast O3 results (to which a water vapor correction has been applied) gives us confidence that water vapor interference is not the source of the observed ozone flux. Second, the conditions tested by Wilson and Birks (2006) involved step changes from 0-90% RH, which are not comparable to those observed during flux legs. In RF03-C and RF06-A, the maximum ΔRH was 20% over the course of several minutes; in other flux legs, e.g. RF07-A, RH varied by no more than ~5%. In all cases, the water vapor concentration changed gradually rather than in "steps". Finally, for the sake of clarity we point out that the method of Wilson and Birks (2006) does not use the Nafion semipermeable membrane to dry the sampling stream. Rather, they use the Nafion membrane to equilibrate the sampling and reference streams. While the method is elegant, we emphasize that it solves a problem that is not relevant to two of the instruments used in this work.

We did consider the effect of changes in water vapor ($dH_2O/dt$) on instrument agreement and find no systematic effect. We have added Figure S2 to the SI text:

[Figure]

Figure S2: Effect of water vapor changes on Fast O3/FAIRO intercomparison. Ratio of Fast O3 to FAIRO 1 is shown as gray crosses. Ratio of Fast O3 to FAIRO 2 is shown as color-coded dots, with black dots indicating low ozone and hotter colors indicating higher ozone (color scale ends at 100 ppbv, yellow). Comparisons are calculated from data averaged over 10 s. No systematic behavior is observed.

Q3: Boylan et al. (2014) dedicated a full manuscript to the study of water vapor interference in eddy correlation ozone flux measurements by chemiluminescence, using an instrument similar to one of the analyzers used in this study. Importantly, they emphasize that the error from the signal quenching is not just affecting the absolute ozone mole fraction result, but that it will bias the ozone flux determination, with the relative error being dependent on the magnitude and the relative ratio of the ozone versus the water flux. These authors present a solution to this problem by drying the sample stream, similar to what Wilson and Birks (2006) proposed in their earlier work. Unfortunately, the important experiments, findings, and recommendations of Boylan et al. (2014) were not considered by Chiu et al.

R3: We respectfully disagree. The original manuscript had evaluated the water sensitivity, and states "Neglecting the water vapor correction altogether decreased the calculated exchange velocity (see Sect. 2.4) by 5%". The reviewer must have missed this in the original manuscript.

Expanding on this, the method of Boylan et al. (2014) decreases the magnitude of the Ridley et al. (1992) water vapor correction but does not obviate the need for such a correction in the first place. Indeed, Boylan et al. state that their work "confirms the correctness of… developed correction algorithms." They calculate a water vapor correction factor $\alpha = 4.15 \times 10^{-3}$, which is within the Ridley et al. (1992) error bars. The original manuscript had also varied the water vapor correction to the full Ridely et al. (1992) range ($4.0\text{-}4.6 \times 10^{-3}$). This changes the ozone flux by only 0.7%. Thus, the Boylan et al. (2014) water vapor correction is already considered in the original manuscript.

Moreover, we point out that Boylan et al. (2014) themselves, referring to the ozone frequency response, state that while the Nafion dryer reduces the water vapor flux by 97%, "the spectral components of the ozone signal remained unchanged." They also conclude, "The ozone mean concentration and ozone fast fluctuations were not affected by the Nafion dryer." Rather, the primary benefit of the Boylan et al. (2014) method is that it simplifies the ozone volume mixing ratio calculation. Thus, we consider the Boylan et al. (2014) method "nice to have," but not critical.

Q4: It is striking that ozone exchange velocities showed a high response to water vapor fluxes. This is exactly the interference that the Boylan et al. paper focuses on. While the three instruments response in a similar direction, this may well be from a similar response to the water vapor interference. Unfortunately, Chiu et al. do not present a convincing case that these ozone fluxes are real and not an interference effect.

A4: We respectfully disagree. First and foremost, we added the following text to the revised manuscript:

The water vapor interference for the coumarin instruments goes in the opposite direction than for the UV instruments, i.e. water vapor makes Fast O3 less sensitive to ozone, but FAIRO more sensitive (Güsten et al., 1992; Schurath et al., 1991; Zahn et al., 2012). The fact that all three instruments agree after water vapor correction gives us confidence that water vapor bias is removed.

Second, RF06-A-1 shows a case in which ozone fluxes are below detection even when water vapor flux is observed, and the fastest ozone exchange velocities do not coincide with the greatest water vapor fluxes. These observations are not consistent with ozone flux being an artifact of water vapor interference.

Third, the water vapor fluxes in RF04-A-1, RF07-A-4, and RF03-A-1 are 1.65, 1.32, and $1.25 \times 10^{15}$ molec $cm^2$ $s^{-1}$, respectively. The corresponding ozone exchange velocities in these legs are $0.036 \pm 0.006$, $-0.033 \pm 0.004$, and $0.024 \pm 0.012$ cm $s^{-1}$, respectively. If ozone flux is purely a water vapor interference artifact, decreasing the water vapor flux from $1.65 \times 10^{15}$ molec $cm^2$ $s^{-1}$ to $1.32 \times 10^{15}$ molec $cm^2$ $s^{-1}$ first changes the direction of the ozone flux from $+0.036$ cm $s^{-1}$ to $-0.033$ cm $s^{-1}$, and further decreasing the water vapor flux from $1.32 \times 10^{15}$ molec $cm^2$ $s^{-1}$ to $1.25 \times 10^{15}$ molec $cm^2$ $s^{-1}$ changes the direction of the ozone flux *again* to $+0.024$ cm $s^{-1}$. This trend is implausible and refutes the hypothesis that ozone flux is a water vapor artifact, especially since the water vapor corrections for the Fast O3 and FAIRO instruments operate in opposite directions.

Finally, as a check, we calculated the temperature flux (which is correlated with the sensible heat flux) as measured by the fast ambient temperature probe, which operates completely independently of water vapor measurements. Below we show Figure 6 with the temperature flux added to the lower right panel as the red trace. The temperature flux shows similar temporal behavior as do the water vapor and ozone fluxes, giving us confidence that we are measuring true atmospheric dynamics, not just water vapor interference.

[Figure]

On the magnitude of possible water vapor interference, we have added the following text to the revised manuscript:

Using the average water vapor concentration during the entire leg for the water vapor correction increases the calculated exchange velocity 2% to 0.134 cm s$^{-1}$; this case represents the extreme case in which water vapor reaching the ozone instruments is completely smeared out by longitudinal diffusion. We conclude that water vapor interference in the Fast O3 instrument contributes at most 5% to the ozone flux uncertainty, and likely less than 2%.

Q5: While this manuscript claims to present an evaluation of three ozone flux techniques, it does not really present a statistical quantitative comparisons and methods evaluation of the ozone fluxes that were determined by the three measurements.

A5: We respectfully disagree, for reasons described above. We refer the reviewer to section 3.4 of the original manuscript.

**Minor issues**

Q6: It is acknowledged that one of the authors is on the editorial board of ATM. For full transparency, the name of the author should be provided.

A6: The phrasing of the COI statement is as prescribed by the AMT submission guidelines. Other papers published in AMT do not single out individuals in similar situations.

**References**

Wilson, K. L. and Birks, J. W.: Mechanism and elimination of a water vapor interference in the measurement of ozone by UV absorbance. Environ. Sci. Technol., 40, 6361–6367, 2006.

Boylan, P. et al.: Characterization and mitigation of water vapor effects in the measurement of ozone by chemiluminescence with nitric oxide. Atmos. Meas. Tech., 7, 1231–1244, 2014.

Clifton, O. et al: Dry Deposition of Ozone Over Land: Processes, measurement, and modeling. Reviews of Geophysics, https://doi.org/10.1029/2019RG000670, 2019.

**Citation**: https://doi.org/10.5194/amt-2023-198-RC2

---

## Author Response (AR1)

Dear Editors:

We are pleased to submit our revised manuscript for our submission amt-2023-198 (Chiu et al., Intercomparison of Fast airborne Ozone Instruments to measure Eddy Covariance Fluxes: Spatial variability in deposition at the ocean surface and evidence for cloud processing). In response to reviewer input, the following significant changes have been made to the manuscript:

1) Concerning possible water vapor interference, Figure S2 has been added to the SI showing that Fast O3 and FAIRO instrument comparisons are independent of changes in water vapor.
2) Discussion of Fast O3 and FAIRO frequency response has been added to Section 3.2.
3) Figure S3 has been added to the SI in support of the revised Section 3.2.
4) Concerning the ozone flux frequency response, Figure S4 has been added to the SI showing cumulative frequency graphs.
5) Discussion of entrainment velocity has been added to Section 3.5 in support of the cloud ozone sink argument.
6) Figure 4 has been modified to include ozone profiles and ascent/descent markers.
7) Figure 6 has been modified with meteorological details (MBL height, cloud layers, and wind barbs) to aid flux interpretations.

In addition, minor grammatical and formatting changes have been made, e.g. introducing figures in order in the text, removing extraneous commas, etc.

Sincerely,

Randall Chiu

---

## Author Response (AR2)

**REVIEW #1**

**Suggestions for revisions:**

The authors have done an admirable job of addressing my comments and I feel both the data quality and case study analysis are now more robust.

However, I have some remaining comments on the spatial heterogeneity case study presented in Figure 7 and the paragraph starting at line 497 in the revised manuscript. It appears this flux leg was flown in a coastal inlet near Anchorage Alaska. It is worth making this clear in the text as spatial heterogeneity in the physical and chemical factors driving ozone deposition in these coastal areas is likely not representative of the open ocean. It is also worth considering that there can be chemical drivers of O3 flux through surface NO emissions which titrate O3 driving an observed negative flux of O3 that is not due to surface deposition. Local shipping emissions could drive this result for example. It would be useful to demonstrate that NO and O3 mixing ratios were roughly constant across this flux leg and are not responsible for the observed variability in the ozone deposition velocity.

Thank you for the suggestions on final revisions. The following text has been added to the manuscript:

NOx data was unavailable during the first of the four segments. However, during the last three segments, NOx concentrations are constant, with NO at $10 \pm 5$ pptv and $NO_2$ at $30 \pm 15$ pptv (note the accuracy of the $NO_2$ sensor is not better than 50 pptv). Unlike the previous case studies presented, RF06-A-1 was flown near the coast of Alaska, where conditions are expected to be different from those over the open ocean. Low NOx variability rules out apparent ozone flux by titration by emissions from urban or shipping sources.

**Citation**: https://doi.org/10.5194/amt-2023-198-RC1